# Hot-Air Drying Significantly Improves the Quality and Functional Activity of Orange Black Tea Compared with Traditional Sunlight Drying

**DOI:** 10.3390/foods12091913

**Published:** 2023-05-07

**Authors:** Zhi Yan, Zhihu Zhou, Yuanfang Jiao, Jiasheng Huang, Zhi Yu, De Zhang, Yuqiong Chen, Dejiang Ni

**Affiliations:** 1National Key Laboratory for Germplasm Innovation and Utilization for Fruit and Vegetable Horticultural Crops, College of Horticulture & Forestry Sciences, Huazhong Agricultural University, Wuhan 430070, China; 2Public Service Center of Quyuan Town, Zigui County, Yichang 443600, China

**Keywords:** orange black tea, processing, drying method, temperature, quality, bioactivity

## Abstract

The quality of traditional sunlight-dried orange black tea can be affected by weather variations, leading to its quality instability. This study investigated the feasibility of replacing sunlight drying with a new hot-air drying method in orange black tea production. The hot-air-dried orange black tea showed better sensory quality than the traditional outdoor-sunlight-dried tea, with a harmonious fruity aroma and sweet–mellow taste. The content of polyphenols and other quality components in the peel and tea leaves was significantly higher after hot-air drying than after sunlight drying. GC-MS analysis showed that the total number of volatile components of hot-air-dried tea (3103.46 μg/g) was higher than that of sunlight-dried tea (3019.19 μg/g). Compared with sunlight-dried orange black tea, the hot-air-dried orange black tea showed higher total antioxidant capacity, with an increase of 21.5% (FRAP), 7.5% (DPPH), and 17.4% (ABTS), as well as an increase of 38.1% and 36.3% in the inhibitory capacity on α-glucosidase and α-amylase activities. Further analysis of the effects of different drying temperatures (40, 45, 50, and 60 °C) on the quality of orange black tea showed that the tea quality gradually decreased with the increase in drying temperature, with the most obvious decrease in the quality of orange black tea at the drying temperature of 60 °C. Low-temperature (40 °C) dried tea had better aroma coordination, higher fruit flavor, greater sweet–mellow taste, and higher retention of functional active substances in orange peel and black tea. In summary, compared with traditional sunlight drying, the hot-air drying method could reduce the drying time from 90 h to 20 h and improve the sensory quality and functional activity of orange black tea, suggesting it can replace the traditional sunlight drying process. This work is significant for improving the quality of orange black tea in practical production.

## 1. Introduction

Black tea is the largest tea category all over the world, with its output accounting for about 60% of the world’s total tea production. Black tea contains a large number of functional active substances, such as tea polyphenols, amino acids, tea polysaccharides, and alkaloids, with the highest content of tea polyphenols, about 10~20% [1]. Additionally, black tea has a variety of health benefits, such as antioxidant, anti-inflammatory, and antitumor functions [2]. Moreover, the polyphenols in black tea were reported to have good antioxidant and anti-inflammatory effects [3].

Citrus peel is rich in flavonoids, hesperidin, limonin, volatile oil, and other bioactive substances [4], which have been confirmed to have antiaging and anti-inflammatory properties and play a role in the prevention and treatment of cancer [5]. For example, citrus peel extract could inhibit the growth of human bladder cancer T24 cells, demonstrating its anti-bladder cancer function to some extent [6]. Additionally, it is reported that the flavonoids in citrus peel were closely related to their anti-inflammatory activity [7].

In China, citrus tea is produced by mixing tea leaves and citrus peel, which was reported to have the function of enhancing human ventilation and digestion [8]. In recent years, citrus peel and flavonoids of Pu’er tea have been shown to have synergistic antioxidant effects based on the study of their antioxidant activity [9]. Additionally, the inhibitory effect of Kumpucha on the proliferation of SGC-7901 and HepG2 tumor cell lines in vitro was found to be significantly stronger than that of Pu’er tea and citrus peel alone, suggesting that flavonoids in citrus peel and phenolic substances in Pu’er tea had synergistic antitumor effects [10]. These reports suggest that the function of citrus black tea is not only determined by the peel and black tea alone but also by the substances in the peel and black tea, thereby producing a synergistic effect of 1 + 1 > 2 [11].

The processing technology of modern citrus black tea generally includes fruit washing, hollowing out the pulp, tea loading, spreading, fixation, and drying, with drying as the longest processing step, which has a great impact on the quality of orange black tea. During drying, the orange peel gradually loses water and forms the appearance quality of green and yellow. Meanwhile, the volatile oil on the peel surface precipitates and adsorbs into the tea leaves, forming the unique aroma quality of citrus black tea, a combination of the aroma of both fruit and tea [12]. As shown in previous studies, the drying method has a great influence on tea quality. For example, a previous study [13] indicated that the content of amino acids and soluble sugars was higher in black tea dried using hot air, and the content of tea polyphenols, catechins, theaflavins, and volatile substances was higher in black tea dried using a microwave, coupled with better taste and aroma. In another study on the influence of different drying methods on camellia quality [14], camellia processed via microwave drying had a more natural appearance, darker color, and higher fragrance. The drying method also affects the quality of citrus peel. Traditionally, citrus peel is usually dried under outdoor sunlight, but changes in weather may cause inconsistent quality. When the temperature is too low, green orange white tea tends to have a sour and astringent taste, even with the occurrence of mildew deterioration [15]. In recent years, other drying methods have also been used to process citrus peel. For example, Ming-Yue compared the difference in sensory quality between hot-air-dried and sunlight-dried citrus peel samples and found that the color of sunlight-dried peel changed greatly and was dark brown, in contrast to the better sensory quality of hot-air-dried peel [16].

“Taoye” sweet orange (*C. sinensis* Osb. cv. “Taoye”) is a local characteristic navel orange species in China. As the peel is sweet, it is often dried and stored by local people, which can be boiled for drinking or stewed with meat [17]. Yichang black tea is one of the main congou black tea varieties in China [18]. In recent decades, orange black tea has been processed using “Taoye” sweet orange peel and black tea, and the product is popular among consumers. However, no research has ever studied the effects of different drying methods on the sensory quality of “Taoye” sweet orange black tea. In this study, the same raw materials of tea and orange were processed into orange black tea following the procedures of washing fruit, removing the pulp, loading tea leaves, spreading, fixation, and drying. By comparing the sensory quality, chemical composition, and functional activity of the orange black tea dried using the new method with the traditional sunlight-dried orange black tea, we sought to improve the process parameters, increase the quality of orange black tea, and enhance the production efficiency, thus promoting the development of citrus tea industry.

## 2. Materials and Methods

### 2.1. Materials and Reagents

The black tea used in this study was a second-class black tea provided by Zigui Yihong Tea Industry Co., Ltd. (Yichang, China). “Taoye” sweet oranges (*C. sinensis* Osb. cv. “Taoye”) were collected on 4 September 2019 from Zigui County, Hubei, China.

Main reagents included anhydrous diethyl ether (chromatographic grade, Korea Doksan Company, Seoul, Republic of Korea); methanol (chromatographic grade, Thermo Fisher Scientific, Waltham, MA, USA); folinol (analytical grade); catechin standard (purity ≥ 98%) (Sigma Company, St. Louis, MI, USA); sodium bicarbonate (analytical grade, Tianjin Ruijinte Chemicals Co., Ltd., Tianjin, China); trichloromethane (chromatographic grade, Tianjin Yongda Chemical Reagent Development Center); ninhydrin; anthrone; potassium dihydrogen phosphate; disodium hydrogen phosphate; stannous chloride; sodium potassium tartrate; folinol; ethanol; methanol; n-butanol; ethyl acetate; oxalic acid; anhydrous sodium sulfate; a cyclohexanone internal standard solution (98% purity, China Pharmaceutical (Group) Shanghai Chemical Reagent Company); Rutin; phosphate; hesperidin; gallic acid; limonin; coomassie brilliant blue g250; bovine serum albumin BSA; α-glucosidase; and α-glucosidase (Shanghai Yuye Biotechnology Co., Ltd., Shanghai, China). A total antioxidant capacity FRAP kit, a total antioxidant capacity DPPH kit, and a total antioxidant capacity ABTS kit (Suzhou Keming Biotechnology Co., Ltd., Suzhou, China) were also used to detect the antioxidant capacity of orange black tea.

### 2.2. Preparation of Orange Black Tea

Orange black tea was prepared as shown in Figure 1. The water content of black tea was about 6%, and the fresh fruit diameter of “Taoye” sweet oranges (*C. sinensis* Osb. cv. “Taoye”) was about 5 cm. First, after water washing and drying at room temperature, the pulp was removed from the “Taoye” sweet oranges (*C. sinensis* Osb. cv. “Taoye”), followed by washing the husks or peels, drying, filling the husks with black tea, and covering them. We then proceeded to the experimental steps of drying methods and drying temperatures. For the analysis of the drying method, the above-prepared orange black tea samples were placed in the room for 5 h, followed by using a 6CZG-60 drying machine (Zhejiang Lvfeng Machinery Co., Ltd., Quzhou, China) for fixation with the temperature and time widely used in current production. Specifically, the samples were fixed by heating at 50 °C for 10 min, 70 °C for 20 min, 85 °C for 10 min, 90 °C for 10 min, and finally cooled to 50 °C for 20 min. After fixation and cooling to room temperature, the samples were moistened for 1 h and then dried using two different methods. One portion of the samples was hot-air-dried by using the above 6CZG-60 box-type drying machine under the conditions of 45 °C for 2 h, 50 °C for 6 h, 80 °C for 10 h, 65 °C for 2 h, and 75 °C for 15 min. The other portion of the samples was dried continuously for 4 days under outdoor sunlight at 35–40 °C during the day and stored at 4 °C at night.

Analysis of different drying temperatures: the peels/husks filled with tea leaves were fixed as described above, followed by drying with the 6CZG-60 box-type drying machine separately at 40 °C for 26 h, 45 °C for 24 h, 50 °C for 22 h, or 60 °C for 20 h.

### 2.3. Analytical Methods

Sensory evaluation: After appearance evaluation, black tea (3 g) and orange peel (1.2 g) were placed into a 250 mL evaluation cup (a cup with a lid). Then, the cup was filled with boiling water, covered with the lid, and soaked for 5 min. Finally, the tea infusion was evaluated in terms of taste, aroma, and color. The sensory evaluation team consisted of 5 members who performed aroma evaluation (in terms of sweet aroma concentration, fruity aroma level, pleasantness, and coordination) and taste assessment (in terms of acidity, sweetness, bitterness, astringency, and coordination) based on the specific scoring standards shown in Appendix A.

Determination of polyphenols [19]: Briefly, 0.2 g sample powder was weighed, followed by the addition of 10 mL 70% methanol and extraction using a water bath at 70 °C, and then the folinol colorimetric method was used to determine the content of polyphenols.

Determination of the content of free amino acids and soluble sugars in tea [12]: Briefly, a 0.5 g sample was weighed and extracted with 50 mL of boiling water. Then, the tea broth was filtered, and the soluble sugars and free amino acids were analyzed using the anthrone–sulfuric acid colorimetric method and hydrin colorimetric method, respectively. 

Detection of orange peel polysaccharide [20]: First, a 0.2 g sample was weighed and supplemented with 40 mL of 80% ethanol, followed by refluxing at 95 °C for 1 h and adding 100 mL of distilled water to the boiling water bath for extraction. Finally, the tea broth was filtered, and the anthrone–sulfuric acid method was used to determine the polysaccharide content. 

Detection of theaflavin, theaflavin, and theaflavin content in tea [21]: First, 3 g sample powder was weighed and extracted with 125 mL of boiling water. After the filtration of tea broth, the systematic detection method was used to determine the content of theaflavin, theaflavin, and theaflavin. 

Determination of orange peel flavonoid content [22]: Briefly, 3 g sample powder was weighed and ultrasonically extracted for 30 min with absolute ethanol. After filtration, the aluminum nitrate method was used to analyze flavonoids. 

Determination of orange peel soluble protein content [23]: Briefly, 3 g sample powder was weighed and extracted for 10 min in a 100 °C boiling water bath. After centrifugation at 3000 rpm for 10 min, the supernatant was collected, and the Coomassie brilliant blue method was used to analyze the soluble protein. 

Simultaneous HPLC determination of hesperidin, synephrine, and limonopterin content in “Taoye” sweet orange peel: Briefly, 0.1 g of ground orange peel powder was weighed and mixed with 10 mL methanol. Then, ultrasonic extraction for 30 min and filtration were performed. After adjusting the volume to 10 mL, 1 mL was passed through 0.22 μm filter membrane for HPLC under the following conditions: Agilent ZORBAX SB-C 18 (250 mm × 4.6 mm × 5 μm) column; 35 °C column temperature; 1 mL·min^−1^ flow rate; 5 μL injection volume; 210 nm and 283 nm detection wavelength; aqueous phosphate pH = 3.7 (mobile phase A); and methanol:acetonitrile = 1:1 (mobile phase B). The elution gradient values are listed in Table 1.

Determination of volatile constituents: Headspace solid phase microextraction (HS-SPME) was employed to extract aroma substances. Briefly, the extraction fibers (DVB/CAR/PDMS) were placed into a GC injector for aging at 250 °C for 30 min. Meanwhile, 1 g of orange black tea sample (black tea and orange peel mixed at 7:3) was weighed and put into a 20 mL headspace bottle and added with a 1 mL cyclohexanone internal standard solution and 5 mL boiling brine [4], followed by inserting the extraction head, heating in a 60 °C water bath for 1 h and analysis with a gas mass spectrometer (DSQ-II, Thermo Fisher Scientific, USA). The chromatographic conditions were as follows: 30 mm × 0.25 mm × 0.22 μm DB-5MS column, 230 °C injection temperature, and 1.0 mL/min column flow rate, high purity (≥99.99%) helium as carrier gas. The initial temperature was set at 45 °C, followed by 7 °C/min to 80 °C, 2 °C/min to 90 °C and holding for 2 min, 3 °C/min to 100 °C and holding for 2 min, 3 °C/min to 130 °C and holding for 2 min, 3 °C/min to 150 °C and holding for 2 min, and finally, 10 °C/min to 230 °C and holding for 5 min; column box temperature was set at 40 °C, with no split injection.

Qualitative and quantitative analysis of volatile components: Each peak was qualitatively analyzed according to the mass spectrometry database and the existing mass spectral identification results, and the ratio of the peak area of each component to the peak of the internal standard was used to calculate the content of the component as the relative content of the internal standard of cyclohexanone.

The antioxidant capacity of orange black tea was detected using ABTS, FRAP, and DPPH kits (Suzhou Keming Biotechnology Co., Ltd., Suzhou, China). Briefly, a 1 g tea sample was extracted for 10 min with 9 mL boiling distilled water. After centrifugation at 3000 rpm for 5 min, the supernatant was collected for analysis [24].

The FRAP method: Following the instructions of the manufacturer, the reagent was prepared, and the mixture (190 μL) was added to the reaction system of 96 microplates. Next, 10 μL of blank and different test samples were added. After standing at room temperature for 20 min, the absorbance value at 593 nm was determined, and the antioxidant capacity was calculated using the following equation: ΔA = A measure − A blank.

The ABTS method: Following the instructions of the manufacturer, the reagent was prepared, and the mixture (190 μL) was added to the reaction system of 96 microplates. Next, 10 μL of blank and different test samples were added. Then, the resultants were left standing at room temperature for 10 min. Next, the absorbance value at 734 nm was measured, and the antioxidant capacity was calculated using the following equation: ΔA = A measure − A blank.

The DPPH method: Following the instructions of the manufacturer, the reagent was prepared, and 380 μL of the working solution was brought to 500 μL volume. Next, 20 μL of blank and different test samples were added. Finally, the absorbance of 200 μL solution was tested at 515 nm in a 96-well plate at room temperature, and the antioxidant capacity was calculated using the following equation: ΔA = A measure − A blank.

Determination of the inhibitory effect on α-glucosidase and α-amylase activities: The inhibitory activity of α-glucosidase was determined following a previous study [23]. Briefly, in the 96-microplate reaction system, 40 μL of 1 unit/mL α-glucosidase solution was mixed with 40 μL of different concentrations of test samples. After the reaction at 37 °C for 10 min, 40 μL of 2.5 mmol/mL p-Nitrophenyl-β-D-Galactopyranoside (pNPG) solution was added, and then the reaction was terminated by adding 120 μL sodium carbonate solution (0.2 mol/L). In the final step, after measuring the absorbance value at 405 nm, the semi-inhibitory concentration (IC50) was calculated using the following equation:Enzyme activity inhibition rate (%) = 1 − (*A*_1_ − *A*_2_)/*A*_0_ × 100
where *A*_0_ is the absorbance value after the reaction between enzyme and substrate; *A*_1_ is the absorbance value after adding the sample; and *A*_2_ is the absorbance value of the sample itself.

The assay of α-amylase activity inhibition was conducted using a Solarbio α-amylase activity detection kit, with slight modifications. Briefly, the control, reaction, inhibition, and background tubes were prepared separately with a control tube containing 550 μL distilled water; a reaction tube containing 200 μL of α-amylase solution (3.3 ug/mL); an inhibition tube containing 150 μL of distilled water + 200 μL of α-amylase solution (3.3 ug/mL) + 200 μL of tea soup; and a background tube containing 350 μL distilled water + 200 μL of tea soup.

After cooling for 3 min, 150 μL of reagent II was added separately into the inhibition tube and the reaction tube and incubated for 5 min in a 40 °C water bath. Next, the four tubes were added separately with 150 μL of reagent I and incubated for 10 min in a 100 °C water bath. After cooling for 5 min, OD values at 540 nm were measured separately for the control, reaction, inhibition, and background tubes.

The α-amylase activity inhibition rate was calculated by the following equation:$$\mathrm{Inhibition}\;\mathrm{rate}\;(\%)=(1-\frac{OD_{inhibition}-OD_{background}}{OD_{reaction}-OD_{control}})\times 100$$

### 2.4. Data Analysis

The means ± standard deviations (SDs) of the three replicates were used to express the results. Statistical analysis was performed using SPSS 25.0 software (IBM, Chicago, IL, USA). A multiple-comparison test was performed using the least significant difference (LSD), and the significant difference between groups was determined at *p* < 0.05.

## 3. Results

### 3.1. Influence of Drying Methods on Sensory Quality of Orange Black Tea

The drying temperature and time used in this experiment are also widely employed in current production techniques, and sunlight drying is the traditional drying method. Table 1 shows the effects of drying methods on the sensory quality of orange black tea, and the drying method was found to mainly affect the aroma and taste of orange black tea, with little influence on its appearance and soup color.

The color appearance was green for orange fresh fruit peel, green and yellow for hot-air-dried orange black tea, and orange–yellow for sunlight-dried orange black tea, indicating that color appearance did not significantly change between fresh fruit peel and hot-air-dried orange black tea, but changed more significantly between fresh fruit peel and sunlight-dried orange black tea.

The aroma of orange black tea was evaluated based on aroma type, intensity, and aroma coordination. Aroma coordination refers to the degree of integration between the aromas of the peel and black tea [25]. There were significant differences in the aroma types of orange black tea processed using the two drying methods. Hot-air-dried orange black tea had a better citrus fruit flavor, with fruit aroma in harmony with the tea aroma. However, the sunlight-dried orange black tea exhibited a higher medicinal fragrance, more harmonious with tea fragrance.

The taste coordination of orange black tea can reflect the degree of integration between peel taste and black tea taste. The five taste indicators were analyzed separately, and the results are shown in Table 2. The drying method was found to have little effect on the sweetness of orange black tea. Meanwhile, the sunlight-dried tea soup was found to be slightly more acidic than the hot-air-dried tea soup, but the latter was more bitter and had better taste coordination than the former.

In terms of the total sensory score, the hot-air-drying treatment scored 72.2, significantly higher than that of the sunlight-drying treatment (68.9). Overall, the sensory evaluation results indicated that hot-air drying can significantly improve the quality of orange black tea compared with traditional sunlight drying.

### 3.2. Comparison of Physical and Chemical Components of Orange Black Tea Treated with Different Drying Methods

The drying method had a significant influence on the physical and chemical composition of the orange black tea peel and tea. As shown in Figure 2A, compared with sunlight-dried peel, hot-air-dried peel had higher contents of polyphenols, polysaccharides, soluble proteins, total flavonoids, hesperidin, synephrine, and limonin, while the sunlight-dried peel had significantly higher polysaccharide content.

The influence of drying methods on the physical and chemical components of orange black tea leaves is shown in Figure 2B. Hot-air-dried tea leaves showed significantly higher contents of tea polyphenols and theaflavins than sunlight-dried leaves, but the latter was significantly higher than the former in the contents of thearubigins and theabrownins. Meanwhile, the two drying methods showed no significant effect on the contents of free amino acids and soluble sugars.

### 3.3. Effects of Drying Methods on Volatile Components of Orange Black Tea

During the sensory evaluation, the hot-air-dried orange black tea had a higher fruit flavor, and the fruit flavor was in harmony with the tea flavor, while the sunlight-dried tea had a distinctive medicinal flavor. In order to explore the effects of different drying methods on volatile components in orange black tea, GC-MS was used to determine the volatile components. In the hot-air-dried and sunlight-dried tea samples, 102 and 101 substances were identified, respectively, with the total number of volatile components being slightly higher in hot-air-dried tea (3103.46 μg/g) than in sunlight-dried tea (3019.19 μg/g).

The effects of different drying methods on the volatile components of orange black tea are shown in Figure 3. Hot-air drying and sunlight drying were found to have no significant difference in the content of volatile components. This result was also observed in black tea processed using different drying methods. The volatile components in hot-air-dried and sunlight-dried black tea were reported to be mainly alcohols, ketones, and esters, and the number of types and the content of each type could not be clearly distinguished [26]. However, the volatile characteristics could be distinguished using principal component analysis (PCA) and cluster analysis (CA).

The OPLS-DA model was used to analyze the distribution of the 104 volatile compounds in hot-air-dried and sunlight-dried orange black tea, and the OPLS-DA model enabled the clear identification of the two differently dried tea samples (R^2^X = 0.784, R^2^Y = 0.999, Q^2^ = 0.988; Figure 4). The reliability of a model is usually evaluated based on the prediction ability parameter Q^2^ and goodness of fit R^2^. In the present study, the replacement test results showed R^2^ = 0.858 and Q^2^ = −0.375, the slope of the regression line of R^2^ and Q^2^ > 1, and the intercept of the Q^2^ regression line was negative, indicating that the model had a good fitting and high predictive ability (Figure 4B). Therefore, the model allowed us to effectively distinguish the two kinds of orange black tea dried using the two different methods.

VIP value can be used to assess the contribution of variables to the model, and variables with VIP > 1 are usually considered important variables, so the larger the VIP value, the more significant the difference in volatile compounds between the two different drying treatments. There were 64 substances with VIP > 1, mainly including alcohols, aldehydes, alkenes, esters, and other substances (Figure 5); thus, significant differences were observed in the volatile components of hot-air-dried and sunlight-dried orange black tea. Specifically, sunlight-dried tea showed higher content than hot-air-dried tea in 34 substances, mainly including 12 kinds of alcohols and 8 kinds of aldehydes, while hot-air-dried tea had a higher content in 30 substances, mainly including 10 kinds of alkenes, 7 kinds of aldehydes, and 7 kinds of alcohols.

The substances contained in the hot-air-dried tea, but not in the sunlight-dried tea, were γ-cineole, methyl jasmonate, and n-hexadecane, while the substances contained in the latter but not in the former were α-pentylcinnamaldehyde and camphor.

### 3.4. Comparison of Functional Activity of Orange Black Tea Treated with Different Drying Methods

The differences in the functional activity of orange black tea dried with different methods are shown in Figure 6. As shown in Figure 6A–C, the hot-air-drying treatment led to significantly higher overall antioxidant activity against FRAP, DPPH, and ABTS than sunlight drying, suggesting that hot-air-dried orange black tea possesses stronger antioxidant ability. In Figure 6D, the IC_50_ values of α-glucosidase and α-amylase were significantly higher for the sunlight-drying treatment than the hot-air-drying treatment, indicating that hot-air-dried tea had a stronger inhibitory effect on the two enzymes.

### 3.5. Correlation between Physical and Chemical Components and Functional Activity of Orange Black Tea

The effects of active ingredients on functional activity were further investigated using the correlation analysis of physical and chemical components and functional activity in orange black tea dried with the two different methods. The results are shown in Table 3. A varying degree of correlation was observed between the physicochemical composition and functional activity of the orange black tea. Tea polyphenols, theaflavin, orange peel polyphenols, flavonoids, hesperidin, synephrine, and limonin were significantly and positively correlated with antioxidant indexes against FRAP, DPPH, and ABTS. Except for theaflavin, hesperidin, and limonin, with a correlation of 0.05 with DPPH and DPPH, the other indexes were significantly correlated with FRAP, DPPH, and ABTS at 0.01 level.

The IC_50_ values of α-glucosidase and α-amylase were negatively correlated with tea polyphenols, theaflavin, hesperidin, hesperidin, synephrine, and limonin, indicating that tea polyphenols, theaflavin, orange peel polyphenols, orange peel polysaccharide, soluble protein, flavonoids, hesperidin, synephrine, and limonin were significantly and positively correlated with the inhibitory ability of orange black tea against α-glucosidase and α-amylase.

### 3.6. Influence of Drying Temperature on Sensory Quality of Orange Black Tea

The above results indicate that hot-air drying is superior to sunlight drying in improving the overall quality of orange black tea, and temperature is the most important process parameter. Thus, the effect of different drying temperatures on the quality of orange black tea was further investigated.

The sensory evaluation results are shown in Table 4. The drying temperature was found to affect the appearance, color, aroma, and taste of orange black tea but had little effect on the tea soup color.

In terms of appearance and color, there was the smallest difference in appearance and color between 40 °C dried orange black tea and fresh fruit peel, which was green, yellow, and bright. However, the color of 60 °C dried orange black tea was yellowish brown with dark brightness, exhibiting the largest color difference between the tea and the fresh fruit peel.

In terms of aroma, with the increase in temperature, the fruit aroma of orange black tea gradually weakened. The fruit aroma was higher in 40 °C dried tea and weaker in 60 °C dried tea, due to exposure of tea aroma and poor aroma coordination. Meanwhile, the drying temperature also had a significant effect on the five taste indexes of orange black tea. With the increase in the drying temperature, the taste of orange black tea gradually turned sour, coupled with a gradual decrease in sweetness, a gradual increase in bitterness and astringency, and thus a gradual decrease in taste coordination. Overall, the sensory score of the orange black tea tended to decrease with the gradual increase in drying temperature. Compared with high-temperature (60 °C) drying, low-temperature (40 °C) drying can significantly improve the quality of orange black tea.

### 3.7. Influence of Drying Temperature on Physical and Chemical Components of Orange Black Tea

The effects of different drying temperatures on the physicochemical composition content of orange black tea peel are shown in Figure 7A. No significant difference was observed in the soluble protein content of the tea peel samples dried at different temperatures. However, with the increase in the drying temperature, the contents of hesperidin, polyphenols, polysaccharides, total flavonoids, synephrine, and limonin significantly decreased in the higher-temperature-dried peel samples. The effects of different drying temperatures on the content of physical and chemical components of orange black tea leaves are shown in Figure 7B. With the increase in the drying temperature, the tea samples’ soluble sugar content showed a significant decrease, whereas the theabrownin content had a significant increase, and there was no significant difference in the content of polyphenols, amino acids, theocyanins, and theaflavins. These results indicate that the drying temperature also affects the content of some substances in orange black tea leaves.

## 4. Discussion

Drying, as the longest processing step of modern citrus black tea, has a great influence on the quality of orange black tea. During drying, the orange peel gradually loses water and forms the appearance quality of green and yellow. Meanwhile, the volatile oil on the peel surface precipitates and adsorbs into tea leaves, forming the unique aroma quality of citrus black tea, a combination of the aroma of both fruit and tea [12]. Traditionally, citrus peel is usually dried under outdoor sunlight, but changes in weather may cause inconsistent quality. When the temperature is too low, the green-orange white tea tends to have a sour and astringent taste, even with the occurrence of mildew deterioration. In recent years, other drying methods have also been used to process citrus peel [15].

The drying method can affect the color, aroma, and taste of orange black tea. This is consistent with several previous studies. For example, the color was found to be quite different between sunlight-dried and hot-air-dried citrus peels, with an orange–yellow and dark color for the sunlight-dried citrus peel [16]. Additionally, in a study on the influence of different drying methods on the color of red dates, the hot-air-dried red dates had the smallest color change and maintained a bright red color, while the sunlight-dried red dates had a larger color change, with a dark red color [27]. Moreover, in a study on tarragon leaves, it was shown that with the increase in the drying temperature (40–90 °C), the color change in tarragon leaves gradually increased, and the original color of tarragon leaves could be better maintained at a lower drying temperature (40 °C) [28].

The drying method can affect the aroma of jujube, as naturally air-dried jujube was found to have lower aroma quality, while hot-air-dried jujube had a higher sweet aroma and better aroma quality, possibly because ultraviolet light in the air promotes the generation of some aroma substances [15]. Many studies have shown that the drying method can affect the taste of food. For example, the drying method was found to have a significant impact on the taste of loquat tea; freeze-dried loquat tea had a sweet and mellow taste [29]. In another study on the effect of different drying methods on the taste of kiwifruit, hot-air-dried kiwifruit had a less salty taste and better taste relative to far-infrared-dried and vacuum-freeze-dried kiwifruit [30]. Moreover, compared with vacuum drying, microwave drying, and natural air drying, hot-air drying could preserve the flavor components of Pleurotus eryngii to the greatest extent, enabling a significantly higher umami flavor than any other drying method [31].

The drying method also had a significant influence on the physical and chemical composition of the orange black tea peel and tea. This is consistent with the results of several previous studies. For instance, under the treatment of hot-air drying, sunlight drying, shade drying, and vacuum drying, hot-air-dried citrus peel showed the highest content of flavonoids [32]. Additionally, in a study on the effect of drying methods on the chemical composition of citrus falling fruit, hot-air drying was found conducive to the retention of flavonoids [33]. Furthermore, high-temperature treatment was reported to increase the contents of total phenols and total flavonoids in orange peel [34], which may be one of the reasons for the high content of total phenols and total flavonoids in hot-air-dried orange black tea peel. During the processing of orange black tea, sunlight drying took a longer time (90 h) than hot-air drying (20 h), causing the oxidization and degradation of more phenolic substances in black tea [35], hence the accumulation of more theabrownins. Additionally, the contents of polyphenols and various catechin components were also found to be higher in hot-air-dried green brick tea than in sunlight-dried green brick tea [35].

During tea processing, olefin can be hydrolyzed, lysed, and released under the action of heat, which has a strong fruity and sweet fragrance [36]. The temperature is higher during hot-air drying, enabling a higher content of olefin in the hot-air-dried orange black tea. We also found that the substances contained in the sunlight-dried tea, but not in the hot-air-dried tea, are α-pentylcinnamaldehyde and camphor. α-Pentylcinnamaldehyde can be synthesized from n-heptanaldehyde and benzaldehyde under certain conditions. α-Pentylcinnamaldehyde has the aroma of jasmine, and camphor has the smell of camphor wood, which may be the reason for the medicinal flavor of sunlight-dried orange black tea [37].

This study revealed the effects of the drying method on the antioxidant activities and α-glucosidase and α-amylase activities of orange black tea. It was found that hot-air drying could improve antioxidant activities and inhibit the α-glucosidase and α-amylase activities of orange black tea. The temperature of hot-air drying (about 80 °C) is higher than that of sunlight drying (about 40 °C), which may contribute to the increase in antioxidant ability. As reported in a previous study [35], with the increase in the drying temperature, the antioxidant capacity of orange peel gradually increased, which may be one of the reasons for the stronger antioxidant capacity of hot-air-dried orange black tea. The drying method was shown to affect the inhibitory activity of α-glucosidase in previous studies. For example, the inhibitory ability of hawthorn to α-glucosidase was found to vary significantly in different drying methods, and hot-air-dried okra was reported to have a stronger inhibitory effect on α-glucosidase and α-amylase than other drying treatments [38].

Tea polyphenols, theaflavin, orange peel polyphenols, flavonoids, hesperidin, synephrine, and limonin were significantly and positively correlated with antioxidant indexes against FRAP, DPPH, and ABTS. There is sufficient evidence that polyphenols, flavonoids, and hesperidin all contain hydroxyl groups, which can easily interact with oxygen and thus have strong antioxidant activities [39,40,41].

Tea polyphenols, theaflavin, orange peel polyphenols, orange peel polysaccharides, soluble proteins, flavonoids, hesperidin, synephrine, and limonin were significantly and positively correlated with the inhibitory ability of orange black tea against α-glucosidase and α-amylase. This is supported by the findings of other studies, where blueberry leaf polyphenols were shown to have stable inhibitory effects on α-amylase and α-glucosidase, probably because both α-amylase and α-glucosidase are all protein structures, and polyphenols can bind proteins through hydrogen bonds, hydrophobic interactions, and covalent bonds [42], thus affecting their structure and properties and then inhibiting enzyme activity.

Drying temperature can affect the quality of orange black tea. This phenomenon has also been found in other food-drying processes. For example, the apricot dried at 40 °C had better aroma quality, with aroma components similar to those of natural apricot and a higher relative content, while the apricot dried at 60 °C had poor aroma quality, with a lower quantity and relative content of aroma substances [43]. Additionally, low-temperature drying (about 40 °C) could retain the citrus fruit flavor better than high-temperature drying, with a higher fruity flavor in the processed orange pucha [44]. Moreover, a previous study on kiwi slice drying showed that when kiwi slices were dried at 40 °C and 60 °C, the taste difference was small between 40 °C dried kiwi slices and fresh kiwi slices, with the sweetest taste and lowest acidity in all the test samples; thus, they had higher sensory quality than 60 °C dried kiwi slices [45].

Drying temperature can also affect the content of substances in orange black tea, probably due to the different heat sensitivity of each substance. Many studies have shown that polyphenols, flavonoids, and other substances can be degraded at high temperatures, such as blueberry polyphenols, which were found to be thermally degraded at high temperatures in a study on the stability and thermal degradation kinetics of blueberry polyphenols [46]. In another study on the effects of different drying methods on the citrus peel quality, freeze-dried citrus peel was found to have the best quality, with higher retention of polyphenols and flavonoids, while the citrus peel dried at a high temperature had a lower content of polyphenols [14,47].

Drying temperature can also affect the physicochemical composition of black tea in orange black tea. This is consistent with a previous study on the effects of different foot fire temperatures on the quality of black tea, which indicated that the higher the foot fire temperature, the lower the contents of tea polyphenols, theaflavins, amino acids, and soluble sugars, and the higher the contents of theocyanins and theaflavins [48]. In another study [13], the researchers found that during high-temperature drying, soluble sugar would react with amino acids, leading to a decrease in tea soluble sugar content but an increase in tea theabrownins. In the present study, during the drying process of orange black tea, most of the black tea leaves were wrapped in the “Taoye” sweet orange peel, thus causing small quality changes, which may be the reason for the insignificant changes in polyphenols, amino acids, theaflavins and theocyanins in the orange black tea leaves.

## 5. Conclusions

In this study, a comparison of the quality of hot-air-dried and sunlight-dried orange black tea indicated that the hot-air-dried orange black tea had a harmonious aroma of fruit and tea, a sweet and mellow taste, and a good sensory quality. Additionally, the content of its physical and chemical components was significantly higher than that of sunlight-dried orange black tea in 10 kinds of alkenes, 7 kinds of aldehydes, and 7 kinds of alcohols. Moreover, hot-air-dried orange black tea exhibited stronger total antioxidant capacity against FRAP, DPPH, and ABTS and an inhibitory effect on α-glucosidase and α-amylase activities than sunlight-dried orange black tea. Furthermore, compared with sunlight drying, hot-air drying has the advantages of short drying time (20 h versus 90 h) and all-weather application, so it can replace the traditional sunlight drying method for processing orange black tea, especially hot-air drying at low temperatures (40 °C). The hot-air drying method presented here can endow orange black tea with satisfactory aroma coordination, high fruit flavor, sweet and mellow taste, and high retention of functional active substances in orange peel and black tea, thus contributing to enhancing the quality of citrus tea in practical production.

## Figures and Tables

**Figure 1 foods-12-01913-f001:**
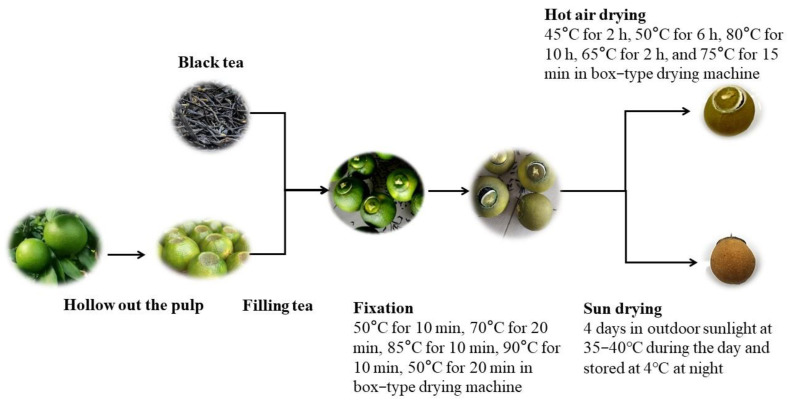
The preparation process of orange black tea.

**Figure 2 foods-12-01913-f002:**
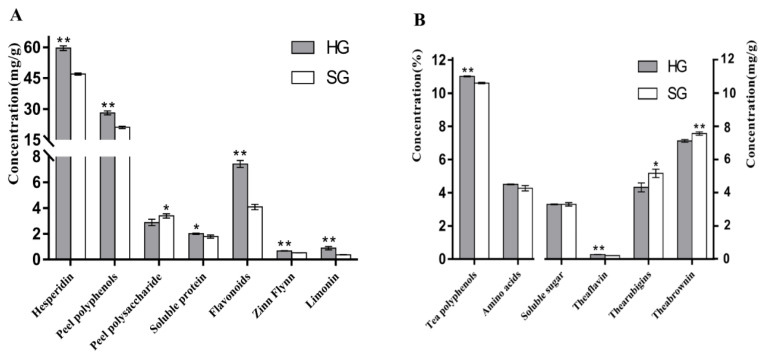
Comparison of physical and chemical components in orange black tea treated using different drying methods: (**A**) comparison of physical and chemical constituents of peel; (**B**) comparison of physical and chemical constituents of tea leaves. HG stands for hot-air drying, and SG stands for sunlight drying. “*” and “**” indicate significant differences at *p* < 0.05 and *p* < 0.01, respectively.

**Figure 3 foods-12-01913-f003:**
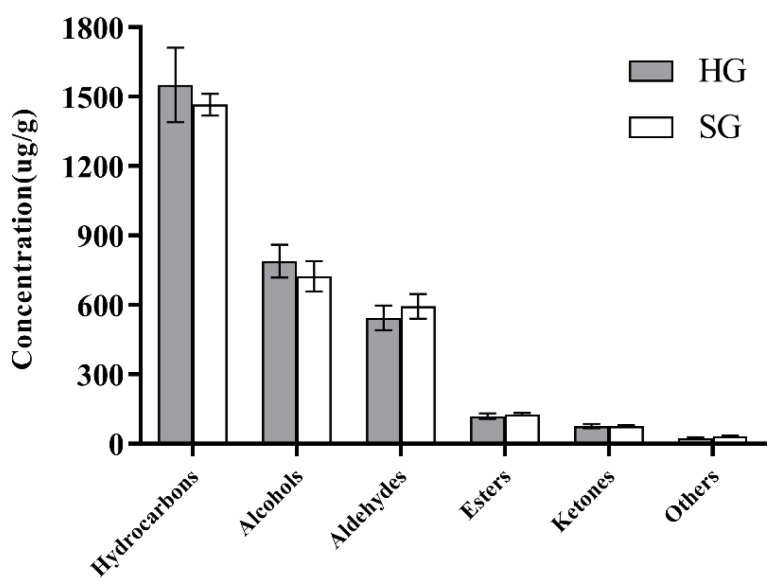
Comparison of volatile components in orange black tea treated using different drying methods. HG and SG stand for hot-air drying and sunlight drying, respectively.

**Figure 4 foods-12-01913-f004:**
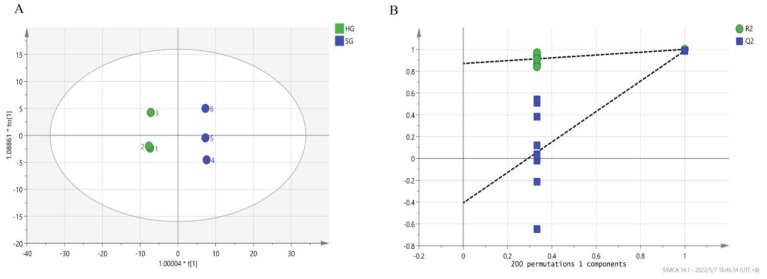
The OPLS-DA plot and cross-validation of orange black tea treated using different drying methods: (**A**) OPLS-DA plot (R^2^X = 0.784, R^2^Y = 0.999, and Q^2^ = 0.988); (**B**) cross-validation of OPLS-DA model with 200 permutation tests (R^2^ = 0.858 and Q^2^ = −0.375). HG and SG stand for hot-air drying and sunlight drying, respectively.

**Figure 5 foods-12-01913-f005:**
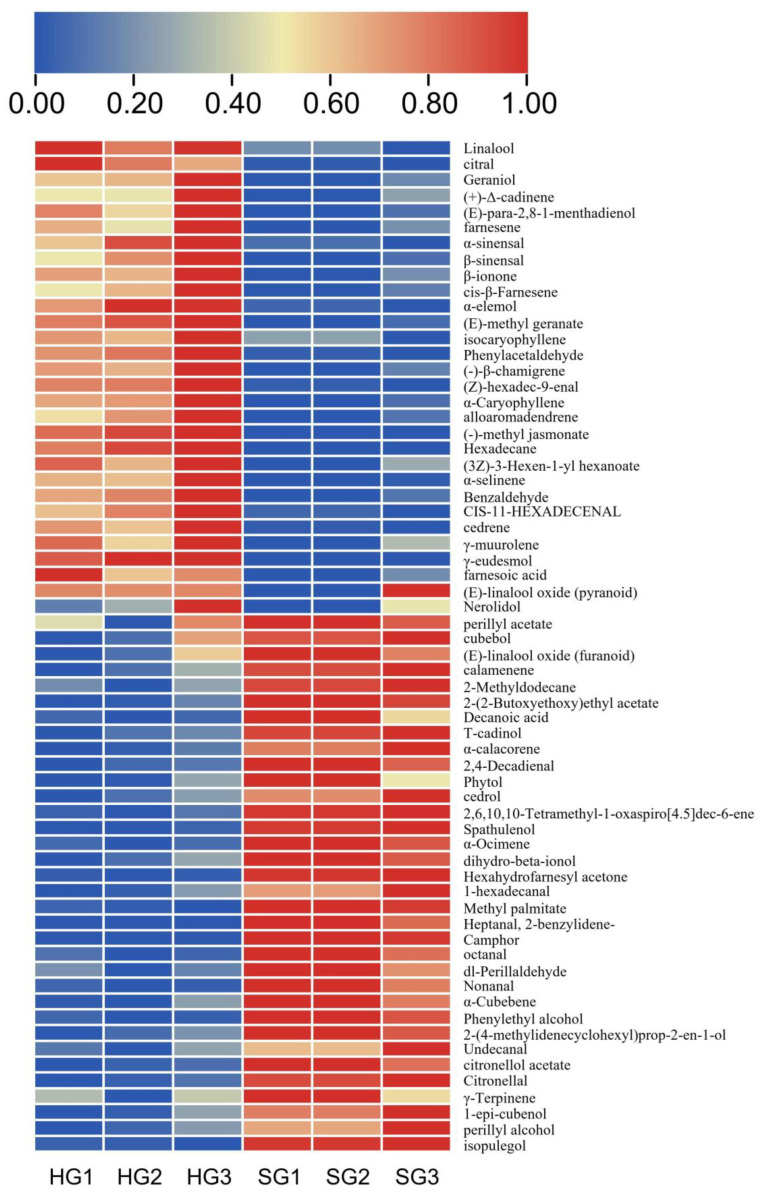
Heat map of 64 substances with VIP > 1 in orange black tea dried using different methods.

**Figure 6 foods-12-01913-f006:**
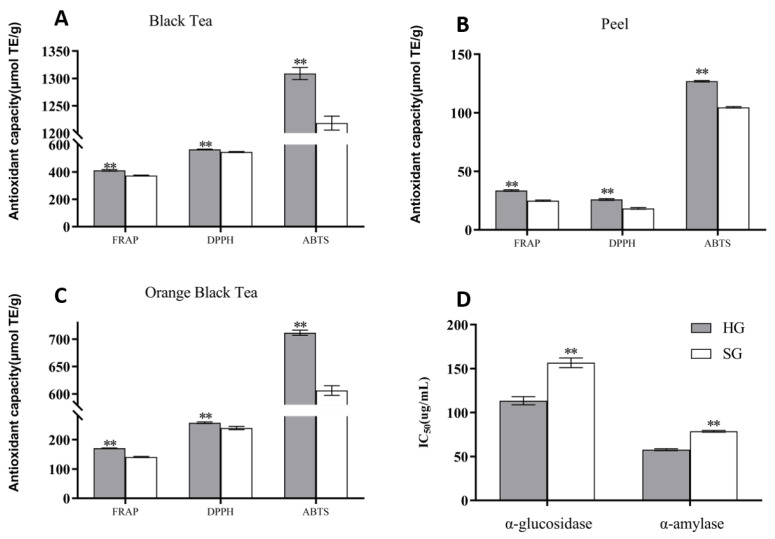
Comparison of functional activities of orange black tea treated with different drying methods: (**A**) comparison of antioxidant activity in tea leaves; (**B**) comparison of antioxidant activity in peel; (**C**) comparison of antioxidant activity in orange black tea; (**D**) the inhibitory effect (IC_50_ values) of differently dried orange black tea on α-glucosidase and α-amylase. HG and SG stand for hot-air drying and sunlight drying, respectively. “**” indicate significant differences at *p* < 0.01.

**Figure 7 foods-12-01913-f007:**
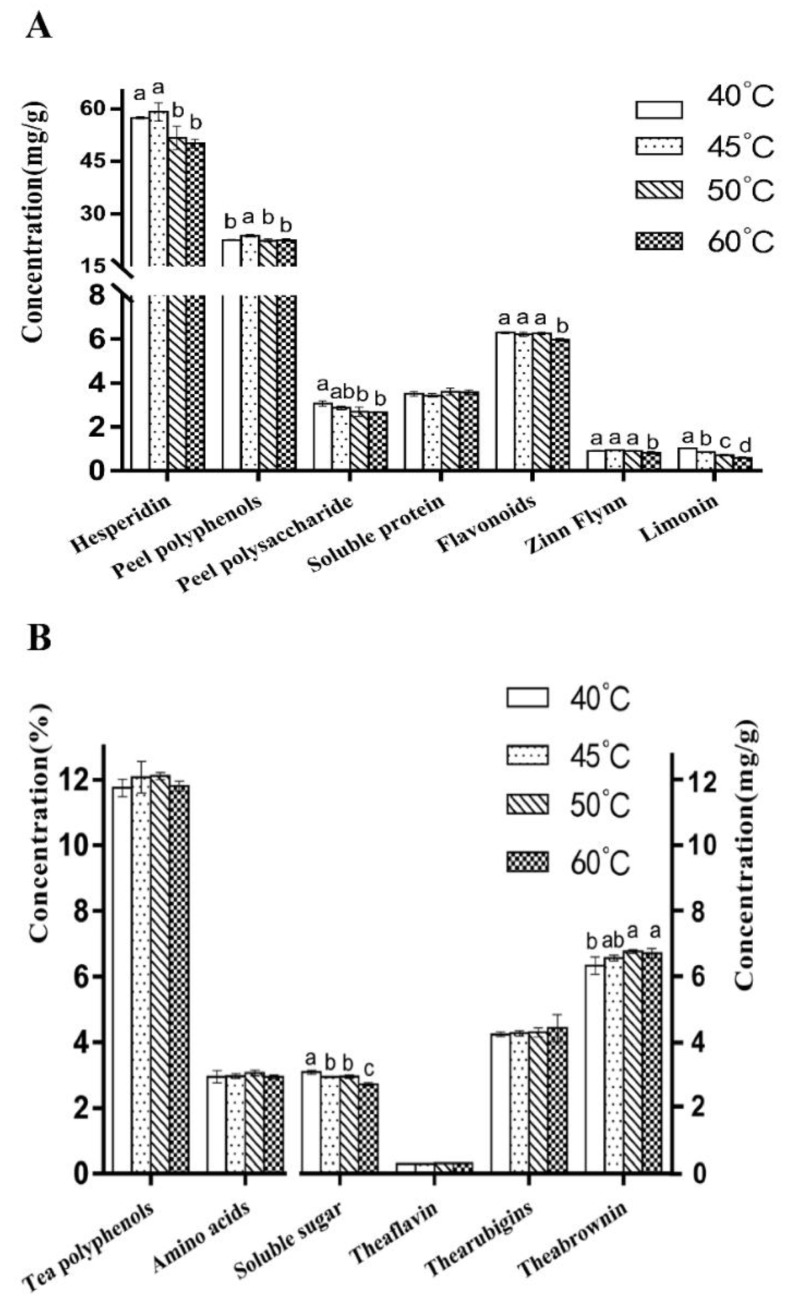
Effects of different drying temperatures on the physical and chemical components of orange black tea: (**A**) comparison of physical and chemical constituents of orange tea peel treated at different drying temperatures; (**B**) comparison of physical and chemical constituents of orange black tea leaves treated at different drying temperatures. Different lowercase letters on the bar indicate significant differences at *p* < 0.05.

**Table 1 foods-12-01913-t001:** HPLC method elution gradient values.

Time (min)	0	5	10	20	25	30	40
Mobile phase A	100	95	75	50	25	5	95
Mobile phase B	0	5	25	50	75	95	5

**Table 2 foods-12-01913-t002:** Sensory evaluation results of orange black tea treated with different drying methods.

Treatment	Appearance	Soup Color	Aroma		Taste					Total Points
			Comment	Aroma Coordination	Acid	Sweet	Bitter	Astringent	Taste Coordination	
Hot-air drying	6.1 ± 0.3 **	9.1 ± 0	14.0 ± 0.1	7.3 ± 0.3	7.1 ± 0.1	7 ± 0.1	7 ± 0 **	8 ± 0.1 **	6.6 ± 0.1 **	72.2 ± 0.3 **
Sun drying	5.0 ± 0.1	9.0 ± 0	15 ± 0.2 **	7.2 ± 0.1	7.2 ± 0.1 **	7 ± 0.2	6.2 ± 0.1	7 ± 0.1	5.3 ± 0.2	68.9 ± 0.2

Note: Each item in the table was evaluated based on the scoring criteria of 10 points for appearance, 10 points for soup color, 15 points for aroma, 15 points for aroma coordination, and 10 points for sour, sweet, bitter, astringent, and coordination. The LSD method was used for pairwise comparison between hot-air drying and sunlight drying. In the same column, “**” indicate significant differences at *p* < 0.01. The sensory evaluation team consisted of 5 members.

**Table 3 foods-12-01913-t003:** Correlation between physical and chemical components and functional activity of orange black tea.

	FRAP	DPPH	ABTS	α-Glucosidase	α-Amylase
Tea polyphenols	0.992 **	0.963 **	0.976 **	−0.986 **	−0.993 **
Amino acids	0.724	0.673	0.817 *	−0.740	−0.764
Soluble sugar	−0.133	−0.235	0.038	0.140	0.080
Theaflavins	0.989 **	0.882 *	0.976 **	−0.958 **	−0.976 **
Thearubigins	−0.916 *	−0.867 *	−0.883 *	0.928 **	0.915 *
Theabrownine	−0.934 **	−0.899 *	−0.943 **	0.921 **	0.938 **
Orange peel polyphenols	0.975 **	0.899 *	0.991 **	−0.969 **	−0.985 **
Orange peel polysaccharide	−0.860 *	−0.739	−0.788	0.817 *	0.817 *
Soluble protein	0.810	0.874 *	0.861 *	−0.881 *	−0.863 *
Flavonoids	0.994 **	0.951 **	0.983 **	−0.989 **	−0.996 **
Hesperidin	0.987 **	0.912 *	0.998 **	−0.982 **	−0.994 **
Synephrine	0.988 **	0.927 **	0.988 **	−0.970 **	−0.989 **
Limonin	0.955 **	0.892 *	0.940 **	−0.912 *	−0.941 **

Note: *, **: significant correlations at 0.05 and 0.01 levels (two-tailed), respectively.

**Table 4 foods-12-01913-t004:** Sensory evaluation results of orange black tea dried at different temperatures.

Drying Temperature	Appearance	Soup Color	Aroma		Taste					Total Points
			Comment	Aroma Coordination	Acid	Sweet	Bitter	Astringent	Taste Coordination	
40 °C	6.6 ± 0.2a	9.2 ± 0	15.3 ± 0.1a	9.0 ± 0a	7.7 ± 0.1a	8.2 ± 0.2a	7.8 ± 0.1a	7.3 ± 0.1a	8.0 ± 0a	79.0 ± 0.3a
45 °C	6.2 ± 0.2a	9.2 ± 0	14.7 ± 0.1b	8.7 ± 0ab	7.3 ± 0.1b	8.0 ± 0.2a	7.3 ± 0.1b	7.0 ± 0.1a	8.0 ± 0a	76.7 ± 0.4b
50 °C	6.0 ± 0.1a	9.2 ± 0	14.5 ± 0.1c	8.5 ± 0b	7.2 ± 0.1b	8.0 ± 0.1a	7.3 ± 0.1b	7.0 ± 0b	7.7 ± 0.1b	75.6 ± 0.2c
60 °C	5.0 ± 0.3b	9.0 ± 0	14.7 ± 0.1b	8.3 ± 0ab	7.0 ± 0.1b	7.5 ± 0.1b	7.3 ± 0.1b	7.0 ± 0b	7.0 ± 0ab	73.0 ± 0.3d

Note: Each item in the table was evaluated based on the scoring criteria of 10 points for appearance, 10 points for soup color, 15 points for aroma, 15 points for aroma coordination, and 10 points for sour, sweet, bitter, astringent, and coordination. The LSD method was used for pairwise comparison. Different lowercase letters in the same column indicate significant differences at *p* < 0.05. The sensory evaluation team consisted of 5 members.

## Data Availability

Data is contained within the article or Appendix A.

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
