# Peer review of "Hot-Air Drying Significantly Improves the Quality and Functional Activity of Orange Black Tea Compared with Traditional Sunlight Drying"

_foods, 2023, doi:10.3390/foods12091913_

Round 1

Reviewer 1 Report

I am very grateful you for the invitation to review manuscript foods-2295441 by Yan and coauthors "Hot air drying significantly improves quality and functional activity of Orange Black Tea relative to traditional sunlight drying”.  This study explored the feasibility of a new hot air drying method in replacing sunlight drying of orange black tea. The work is interesting but needs adjustments to increase the quality of the material.

Comments:

- Abstract: The authors must present the problems related to traditional drying methods and the motivation for carrying out the study. This is not clear in the abstract.

- Please indicate in the abstract a brief and better step-by-step about the work and the evaluated parameters.

- Lines 21-22: The presented difference does not seem so great to justify a method with higher energy expenditure. All issues should be explored for justification.

- Lines 22-23: Please include numerical data referring to the indicated parameters.

- Lines 26-28: Indicate the values, mainly time, in relation to the traditional method.

- Line 29: Change the repeated keywords by different words from the title.

- Line 32: Include numerical data about the market and concentrations of the main components.

- Line 45: Check “Pu 'er tea”. Is this correct?

- Line 67-68: Little negative information is presented regarding traditional drying. Authors should deepen the discussion about the method and its undesirable changes.

-  Line 87: Include the objective at the end of the introduction.

- Lines 90-91: Check the sentence. Some information is missing.

- 2.1. Materials and reagents: Include basic raw material information (moisture and other specifications provided by the manufacturer).

- Figure 1: should be improved by inserting technical and operational information used in the process.

- Lines 108, 109, 119: Does the term “drying” refer to hand drying?

- Line 110: Check the sentence. Some information is missing.

- 2.2. Preparation of orange black tea: Specify how moisture monitoring and completion of the drying process were done.

- Lines 127-130: More information regarding sensory analysis should be added (conditions, number of evaluators, among others).

- Standardize the use of units throughout the text (Ex: ml; mL among others).

- Lines 182-200: Appropriate references must be inserted.

- Lines 242-247: The authors use discussion in the item results. I suggest the unification of the items or adequate separation between material and methods.

- Lines 249-245: See previous comment.

- Lines 256-258; 262-263: This information must also appear in the methodology item since it is not previously detailed.

- Lines 263-266: The authors use discussion in the item results. I suggest the unification of the items or adequate separation between material and methods.

- Line 298: Change “102 and 101 substances” to “102 and 101 substances, respectively”.

- Lines 335-338: The authors use discussion in the item results. I suggest the unification of the items or adequate separation between material and methods.

- Lines 341-344: The authors use discussion in the item results. I suggest the unification of the items or adequate separation between material and methods.

- Lines 382-385: Sentence can be removed as it is repetition of previous sentences.

- Lines 399-405: The authors use discussion in the item results. I suggest the unification of the items or adequate separation between material and methods.

- Lines 430-436: The authors use discussion in the item results. I suggest the unification of the items or adequate separation between material and methods.

- Lines 442-452: The authors use discussion in the item results. I suggest the unification of the items or adequate separation between material and methods.

- 4. Discussion: Authors must change according to previous comments and add the discussion inappropriately inserted in the results item.

- 4. Discussion: Include a discussion of process yield using both methods.

- 4. Discussion: The negative aspects of the traditional method must be highlighted.

Author Response

Dear editor and reviewers:

We sincerely thank you and the reviewers for the valuable comments and suggestions on how to improve the quality of our manuscript. We have carefully read the comments and revised the manuscript as suggested. The responses to all the comments are listed below one by one in bold, and changes in the revised manuscript are highlighted in red.

Responses to Reviewers

To Reviewer #1:

  1. - Abstract: The authors must present the problems related to traditional drying methods and the motivation for carrying out the study. This is not clear in the abstract.

Answer: Thank you for the suggestions. We have added the related information in the abstract.

The quality of traditional sunlight dried orange black tea can be affected by weather variations, leading to its quality instability”

  1. - Please indicate in the abstract a brief and better step-by-step about the work and the evaluated parameters

Answer: Thank you for the suggestions. We have revised the abstract as suggested.

  1. - Lines 21-22: The presented difference does not seem so great to justify a method with higher energy expenditure. All issues should be explored for justification.

Answer: Thanks for the suggestion. The GC-MS results were only a part of our overall analysis, and we comprehensively analyzed the differences between them from the perspectives of sensory quality, chemical composition, and functional activity. In addition, the dryer uses electric heating operation, with an energy consumption of about 1.5 kW/h, and hot air drying can save about 3/4 of the time, reducing the drying time from 90 hours to 20 hours, thus greatly improving the production efficiency.

  1. - Lines 22-23: Please include numerical data referring to the indicated parameters.

Answer: Thank you for the suggestions. We have added the numerical data in the abstract as suggested.

  1. - Lines 26-28: Indicate the values, mainly time, in relation to the traditional method.

Answer: Thank you for the suggestions. We have added the related information in the article:the hot air drying method could reduce the drying time from 90 hours to 20 hours”

  1. - Line 29: Change the repeated keywords by different words from the title.

Answer: Thank you for the suggestions. We have modified the key words to avoid the repetition of the words in the title

  1. - Line 32: Include numerical data about the market and concentrations of the main components.

Answer: Thank you for the suggestions. We have included the numerical data about the market and concentrations of the main components in the revised manuscript as suggested: Black tea is the largest tea category all over the world, with its output accounting for about 60% of the world's total tea production. Black tea contains a large number of functional active substances, such as tea polyphenols, amino acids, tea polysaccharides, and alkaloids, with the highest content of tea polyphenols, about 10%~20%.

  1. - Line 45: Check “Pu'er tea”. Is this correct?

Answer: Thank you for the suggestion. "Pu'er tea" is the standard term or spelling for this type of tea. It is correct.

  1. - Line 67-68: Little negative information is presented regarding traditional drying. Authors should deepen the discussion about the method and its undesirable changes.

Answer: Thank you for the suggestions. We have added the disadvantages the traditional drying method in the revised manuscript.

  1. -  Line 87: Include the objective at the end of the introduction.

Answer: Thank you for the suggestions. We have added the objective in the revised manuscript: By comparing the sensory quality, chemical composition and functional activity of the new method-dried orange black tea and traditional sunlight-dried orange black tea, we intended to improve the process parameters, increase the quality of orange black tea, and enhance the production efficiency, thus promoting the development of citrus tea in-dustry.

  1. - Lines 90-91: Check the sentence. Some information is missing.

Answer: Thank you for the suggestions. We have added the information in in the revised manuscript.

  1. - 2.1. Materials and reagents: Include basic raw material information (moisture and other specifications provided by the manufacturer).

Answer: Thank you for the suggestions. We have added the information in in the revised manuscript: The water content of black tea was about 6%, and the fresh fruit diameter of peach leaf orange was about 5 cm.

  1. - Figure 1: should be improved by inserting technical and operational information used in the process.

Answer: Thank you for the suggestions. We have added to the processing flow chart the technical and operational information used in the process as suggested.

Figure 1. Preparation process of orange black tea.

  1. - Lines 108, 109, 119: Does the term “drying” refer to hand drying?

Answer: Thank you for the suggestions. The term ‘drying’ in line 108 and 109 refer to natural or sunlight drying, and the term ‘drying’ in line 119 refers to drying in a 6CZG-60 box-type drying machine.

  1. - Line 110: Check the sentence. Some information is missing.

Answer: Thank you for the suggestions. We have added the related information in the revised manuscript.

  1. - 2.2. Preparation of orange black tea: Specify how moisture monitoring and completion of the drying process were done.

Answer: Thank you for the suggestions. The drying of orange black tea was performed in a box-type drying machine. After setting the parameter manually, the machine can automatically adjust the temperature and time. After drying for 20 hours, the sample was taken to determine the weight, and after further drying for 1 hour, the weight remained unchanged, which was regarded as the completion of drying.

  1. - Lines 127-130: More information regarding sensory analysis should be added (conditions, number of evaluators, among others).

Answer: Thank you for the suggestions. Details of the sensory review have been added in the revised manuscript (line 143-147): The sensory evaluation team consisted of 5 members, who performed aroma evaluation (in terms of sweet aroma concentration, fruity aroma level, pleasantness, and coordination) and taste assessment (in terms of acidity, sweetness, bitterness, astringency, and coordination) based on the specific scoring standards shown in Supplementary Mate-rials Table S1.

  1. - Standardize the use of units throughout the text (Ex: ml; mL among others).

Answer: Thank you for the suggestions. "ml" has all been replaced with "mL"

  1. - Lines 182-200: Appropriate references must be inserted.

Answer: Thank you for the suggestions. All the methods are based on the biological kit instruction manual, and the corresponding references have been inserted as suggested in the revised manuscript.

  1. - Lines 242-247: The authors use discussion in the item results. I suggest the unification of the items or adequate separation between material and methods.

Answer: Thank you for the suggestions. We have shifted the discussion from the item results to the discussion section in the revised manuscript.

  1. - Lines 249-245: See previous comment.

Answer: Thank you for the suggestions. We have shifted the discussion from the item results to the discussion section in the revised manuscript.

  1. - Lines 256-258; 262-263: This information must also appear in the methodology item since it is not previously detailed.

Answer: Thank you for the suggestions. Sensory review methods have been described in detail in 2.3. Analytical methods in the revised manuscript.

  1. - Lines 263-266: The authors use discussion in the item results. I suggest the unification of the items or adequate separation between material and methods.

Answer: Thank you for the suggestions. We have shifted the discussion from the item results to the discussion section in the revised manuscript.

  1. - Line 298: Change “102 and 101 substances” to “102 and 101 substances, respectively”.

Answer: Thank you for the suggestions. “102 and 101 substances” had been changed to “102 and 101 substances, respectively” as suggested.

  1. - Lines 335-338: The authors use discussion in the item results. I suggest the unification of the items or adequate separation between material and methods.

Answer: Thank you for the suggestions. We have shifted the discussion from the item results to the discussion section in the revised manuscript.

  1. - Lines 341-344: The authors use discussion in the item results. I suggest the unification of the items or adequate separation between material and methods.

Answer: Thank you for the suggestions. We have shifted the discussion from the item results to the discussion section in the revised manuscript.

  1. - Lines 382-385: Sentence can be removed as it is repetition of previous sentences.

Answer: Thank you for the suggestions. We have deleted this sentence as you suggested

  1. - Lines 399-405: The authors use discussion in the item results. I suggest the unification of the items or adequate separation between material and methods.

Answer: Thank you for the suggestions. We have shifted the discussion from the item results to the discussion section in the revised manuscript.

  1. - Lines 430-436: The authors use discussion in the item results. I suggest the unification of the items or adequate separation between material and methods.

Answer: Thank you for the suggestions. We have shifted the discussion from the item results to the discussion section in the revised manuscript.

  1. - Lines 442-452: The authors use discussion in the item results. I suggest the unification of the items or adequate separation between material and methods.

Answer: Thank you for the suggestions. We have shifted the discussion from the item results to the discussion section in the revised manuscript.

  1. - 4. Discussion: Authors must change according to previous comments and add the discussion inappropriately inserted in the results item.

Answer: Thank you for the suggestions. We have shifted the discussion from the item results to the discussion section in the revised manuscript.

  1. - 4. Discussion: Include a discussion of process yield using both methods.

Answer: Thank you for the suggestions. As we have described in detail in the test method, the hot air drying time is about 20 hours, and the sun-light drying time is about 4 days (90 hours). The time difference is so great, indicating the hot air-drying method has higher processing efficiency and higher yield.

  1. - 4. Discussion: The negative aspects of the traditional method must be highlighted.

Answer: Thank you for the suggestions. We highlighted the negative aspects of traditional drying (Line 447-450): Traditionally, citrus peel was usually dried under outdoor sunlight, but weather changes could induce unstable quality. When the temperature is too low, the green orange white tea tends to have a sour and astringent taste, even with the occurrence of mildew deterioration.

Reviewer 2 Report

The manuscript entitled 'Hot air drying significantly improves quality and functional activity of Orange Black Tea relative to traditional sunlight drying' is of good quality and supported by the data. The work is interesting but needs to be corrected.

Below is a list of suggestions that I think would help improve the manuscript.

1.         Please add the Latin (scientific) name of the black tea in the material section

2.         Methods are explained clearly and in detail

3.         Figure 1 (especially 'black tea' should be clearer), please change the photo

4.         The section on conclusions should be better described

5.         The English spelling and grammar of the manuscript have been carefully checked

6.         Please check the style of the references

Author Response

Dear editor and reviewers:

We sincerely thank you and the reviewers for the valuable comments and suggestions on how to improve the quality of our manuscript. We have carefully read the comments and revised the manuscript as suggested. The responses to all the comments are listed below one by one in bold, and changes in the revised manuscript are highlighted in red.

To Reviewer #2:

The manuscript entitled 'Hot air drying significantly improves quality and functional activity of Orange Black Tea relative to traditional sunlight drying' is of good quality and supported by the data. The work is interesting but needs to be corrected.

Below is a list of suggestions that I think would help improve the manuscript.

  1. Please add the Latin (scientific) name of the black tea in the material section

Answer: Thank you for the suggestions. Black tea is Latin for black tea.

  1. Methods are explained clearly and in detail

Answer: Thank you for the suggestions. Some information about the the method has been added in supplementary information

  1. Figure 1 (especially 'black tea' should be clearer), please change the photo.

Answer: Thank you for the suggestions. We have changed to a clearer picture for black tea.

  1. The section on conclusions should be better described

Answer: Thank you for the suggestion. We have rephrased the description of conclusion.

  1. The English spelling and grammar of the manuscript have been carefully checked.

Answer: Thank you for the suggestions. We have checked the grammar and spelling throughout the manuscript as suggested.

  1. Please check the style of the references

Answer: Thank you for the suggestions. We have carefully checked and revised the format of the references.

We tried our best to improve the manuscript and made some changes in the manuscript. These changes will not influence the content and framework of the paper.

We appreciate for Editors/Reviewers’ warm work earnestly, and hope that the correction will meet with approval.

Once again, thank you very much for your comments and suggestions.

Thank you and best regards.

Yours sincerely

Dejiang Ni

Corresponding author: Dejiang Ni
